# Skin Barrier Reinforcement Effect Assessment of a Spot-on Based on Natural Ingredients in a Dog Model of Tape Stripping

**DOI:** 10.3390/vetsci9080390

**Published:** 2022-07-28

**Authors:** Adrien Idée, Marion Mosca, Didier Pin

**Affiliations:** University of Lyon, VetAgro Sup, Interaction Cellule Environnement, 9280 Marcy l’Etoile, France; adrien.idee@vetagro-sup.fr (A.I.); marion.mosca@vetagro-sup.fr (M.M.)

**Keywords:** dog, skin barrier, atopic dermatitis, therapy, transepidermal water loss, hydration

## Abstract

**Simple Summary:**

Canine atopic dermatitis is a disease of dogs in which the skin becomes permeable. Part of treating canine atopic dermatitis involves restoring the skin barrier function; however, few effective therapeutic options exist. The goal of this study was to evaluate the effect of a spot-on containing fatty acids, ceramides and essential oils on two parameters to evaluate the skin barrier function of dogs. We found that this spot-on had a protective effect on the skin barrier function for both parameters. This study suggests that the investigated product may be useful as an adjunctive treatment for canine atopic dermatitis.

**Abstract:**

Skin barrier restoration is an important part of atopic dermatitis therapy. We investigated the effect of a spot-on containing plant-based essential fatty acids and essential oils on skin barrier parameters in a dog model of acute skin barrier disruption, using five healthy beagle dogs maintained in a laboratory setting. Four test sites on the dorsum and a control site on the abdomen were defined on each dog. Transepidermal water loss (TEWL) and skin surface hydration (SSH) were measured before and after tape stripping on the first day and then for three consecutive days, over four consecutive weeks. The spot-on was applied at the end of each of the first three weeks. The increase in TEWL after tape stripping was reduced after the spot-on application and reached control values in Weeks 3 and 4. SSH after tape stripping was reduced in Week 4 compared with the baseline. Thus, the ATOP 7^®^ spot-on significantly reduced acute skin barrier impairment in a dog model. The use of this product should be further evaluated as a potential treatment for skin barrier defects such as canine atopic dermatitis.

## 1. Introduction

The skin’s primary function is to form a barrier that protects against noxious environmental stress, physical and chemical damage, infections and allergens. The skin barrier is composed of host cells (mainly keratinocytes); proteins, such as junction proteins and moisturizing factors; lipids, organized in structured lamellae containing cholesterol, free fatty acids and ceramides; and the skin’s microbiota [1,2].

Transepidermal water loss (TEWL) and skin surface hydration (SSH) have been widely used to assess skin barrier function, especially in atopic dermatitis, a frequent dermatosis associated with lipid, protein and cellular abnormalities. Atopic dermatitis can lead to a dysfunctional skin barrier, allowing the penetration of and sensitization to allergens, as well as increased microbial adherence [3,4]. The resulting inflammation causes a further weakening of skin barrier function, leading to a vicious circle [5]. TEWL measures the amount of water emanating from the skin by using a humidity captor to obtain an evaporation rate [6]. Skin surface hydration (SSH) is a parameter that uses impedance measurements to assess the water content of the skin surface [7]. TEWL and SSH values increase with skin barrier damage. SSH also increases after moisturizer application, since moisturizers augment water content in the stratum corneum. Other less widely used methods to evaluate skin barrier function include pH, modified corneocyte surface area measurements, the evaluation of lipid lamellae organization through electronic microscopy and lipid composition analysis [7,8].

Tape stripping is a minimally invasive method that removes layers from the stratum corneum via successive applications of acetate tape on the skin [9]. This method has been used in dogs. The level of skin disruption is correlated with skin barrier parameters [6] and allows the monitoring of skin barrier restoration over time [10].

Although the skin barrier has been extensively studied in dogs, few treatments effectively improve the skin barrier function both experimentally and clinically in this species [11,12,13,14,15]. The spot-on assessed in this study (ATOP 7^®^ spot-on, Dermoscent^®^, Castres, France) has been used as an adjunctive treatment to lokivetmab in a clinical study of dogs with atopic dermatitis [16]. It contains essential fatty acids, ceramides and essential oils from numerous plants, including turmeric and wintergreen.

The objective of this study was to evaluate the effect of ATOP 7^®^ spot-on (Dermoscent^®^, Castres, France) on TEWL and SSH in a dog model of acute skin barrier disruption. 

## 2. Materials and Methods

All procedures were approved by the Institutional Animal Care and Use Committee of our institute (Ethics Committee reference number 1734).

The study included five healthy male beagle dogs, aged from two to four years, and without any history of dermatological disease or sign of pruritus.

### 2.1. Study Timeline

The study was conducted over four consecutive weeks. Each week (Week 1, Week 2, Week 3, and Week 4) was divided into T0 (before any intervention except clipping), D1 (one hour after tape stripping), D2 (24 h after tape stripping), D3 (48 h after tape stripping) and D4 (72 h after tape stripping). The study protocol is synthesized in Figure 1.

### 2.2. Site Preparation

Four test sites of 10 cm^2^ each (“Site 1”, “Site 2”, “Site 3” and “Site 4”) were demarcated on the back of each dog in the paralumbar region. A spot-on application site (“spot-on”, ATOP 7^®^ spot-on, Dermoscent^®^, Castres, France) was defined at the center of the four test sites, equidistant to the center of each site (Figure 1). A fifth site was defined on the ventral abdomen of each dog and served as a control site (“control”). Hair was gently but closely clipped at each site with a 0.5 mm blade.

Each week, a single dorsal site was assigned as the test site (e.g., Site 1 for Week 1, Site 2 for Week 2, etc.). The ventral control site (“control”) was the same for the duration of the experiment. Tape stripping was performed on the designated test site on the first day of each week. On Week 1, before any spot-on application, tape stripping was performed on Site 1 until the skin was shiny. The number of tape strips required varied from 16 to 22. The same number of tape strips was repeated for a given dog over the three following weeks. On the last day (D4) of Week 1, Week 2 and Week 3, a spot-on was applied at the “spot-on” site, for a total of three successive weekly applications. Site 1 was therefore assigned as the baseline site, before spot-on applications, whereas Site 2, Site 3 and Site 4 represented test sites after one, two and three spot-on applications, respectively.

### 2.3. Measurements

TEWL was measured using an AquaFlux™ AF200 (Biox Systems, London, UK). SSH was measured using a Corneometer^®^ CM 825 (Courage + Khazaka electronic GmbH, Cologne, Germany), with results expressed in arbitrary units as chosen by the manufacturer. Dogs remained in the test room with controlled conditions (temperature 22 °C, humidity 35%) for at least 30 min before measurements. Three successive measurements were performed and averaged at each point to reduce intraoperator variability. Each week, TEWL and SSH were measured at both the test site and the control site on Day 1 before tape stripping (“T0”), 1 h after tape stripping (“D1”) and then on Day 2 (“D2”), Day 3 (“D3”) and Day 4 (“D4”).

Adverse events were noted if present.

### 2.4. Statistical Analysis

Statistical analysis was performed using GraphPad Prism 9.0.0 and *p*-values less than or equal to 0.05 were considered significant. Shapiro–Wilk tests were used for normality testing and paired t-tests were performed to evaluate the difference in means between the control and test sites or between different time points at the same site.

## 3. Results

### 3.1. Transepidermal Water Loss

The mean TEWL values of all sites before tape stripping (T0) were not significantly different from those of the control site for any week (Figure 2, *p* > 0.05).

The mean TEWL values significantly increased after tape stripping at Site 1 (Week 1) versus the control on D1 (Figure 2a, 77.6 vs. 20.46 g∙m^−2^∙h^−1^; *p* = 0.004) and D2 (68.7 vs. 25.5 g∙m^−2^∙h^−1^; *p* = 0.0009). The elevated TEWL values progressively decreased on D3 (68.3 vs. 35.7 g∙m^−2^∙h^−1^; *p* > 0.05) and D4 (54.6 vs. 29.0 g∙m^−2^∙h^−1^; *p* > 0.05), compared with the control. For Site 2 (Week 2), after the same number of tape strips and one spot-on application, similar results were obtained at the test site and control site, respectively (Figure 2b, D1: 78.0 vs. 25.9 g∙m^−2^∙h^−1^, *p* = 0.003; D2: 73.4 vs. 28.4 g∙m^−2^∙h^−1^, *p* = 0.02; D3: 57.0 vs. 25.9 g∙m^−2^∙h^−1^, *p* > 0.05; D4: 47.2 vs. 24.6 g∙m^−2^∙h^−1^, *p* > 0.05). For Site 3 (Week 3) and Site 4 (Week 4), after the same number of tape strips but two and three spot-on applications respectively, no data points were statistically different from the control (Figure 2c,d).

The mean TEWL values after tape stripping (D1) were used to compare the maximum increase in TEWL for each site. The mean TEWL values of Site 1 on D1 and Site 2 on D1 were not significantly different (*p* > 0.05). In contrast, the mean TEWL values for Site 3 on D1 (*p* = 0.009) and Site 4 on D1 (*p* = 0.0006) were significantly different from Site 1 on D1 (Figure 3).

### 3.2. Skin Surface Hydration

Although the differences were not statistically significant compared with the control site, the mean SSH values increased after tape stripping on D1 for each site (Site 1: 39.3 vs. 16.7, Site 2: 36.2 vs. 11.5; Site 3: 17.2 vs. 9.6; Site 4: 10.26 vs. 7.1, Figure 4a–d), since the measurements were performed deeper in the epidermis. The mean SSH values progressively decreased with each passing day, reflecting skin barrier restoration.

The mean SSH values on D1 did not significantly decrease in Site 2 or Site 3 compared with Site 1, but they were significantly reduced in Site 4 compared with Site 1 (*p* = 0.035, Figure 5). This result correlated with the TEWL values.

### 3.3. Adverse Effects

No adverse effects were observed during the experiment.

## 4. Discussion

This study assessed the effects of a plant-based spot-on containing essential oils, essential fatty acids and ceramides in an acute canine skin barrier impairment model. Three weekly spot-on applications did not induce any observable adverse effects, in accordance with a previous clinical study with no reported adverse effects after one month of four weekly applications [16].

Our model, as previously described [10], simulated acute skin barrier disruptions. Our results seemed to confirm the validity of this model, with an acute and strong increase in TEWL and a non-significant increase in SSH on D1 after tape stripping. Furthermore, these parameters returned to baseline on the fourth day after tape stripping. While this model did not use dogs suffering from canine atopic dermatitis, it did simulate the skin barrier defects observed in this disease.

TEWL and SSH were chosen because they are non-invasive, rapid and repeatable methods that are widely used in skin barrier function studies. TEWL has been extensively measured with a closed-chamber Vapometer (Delfin©, Delfin Technologies Ltd., Kuopio, Finland, [12,14,17]), but a recent study highlighted the high intra- and interobserver variability of this device [7]. We therefore decided to use an AquaFlux™ AF200 (Biox Systems), a different device based on the same principles. This device uses refrigeration to continuously evaporate vapor, allowing for continuous measurements. Although this device has not been validated, we observed a low standard deviation for control values (SD = 4.5; mean = 25.3), suggesting low intraobserver variability. For SSH measurements, our system has been shown to have high repeatability and low intraobserver variability [7].

Two or three successive weekly applications of the spot-on significantly reduced the effect of tape stripping on TEWL and SSH, respectively, as shown by the absence of significantly different values compared with the control site at those time points. These results suggest a skin barrier function improvement and protection after using ATOP 7^®^ spot-on. 

Skin barrier restoration is one of the therapeutic cornerstones for canine atopic dermatitis, as skin barrier dysfunction seems to be at the root of the pathogenesis of this condition [18,19,20]. In previous studies on atopic dermatitis, a spot-on containing fatty acids and cholesterol produced inconsistent results [14,21,22,23], whereas a spot-on containing essential fatty acids and essential oils seemed to provide more consistent results [24,25]. Furthermore, oral supplementation with essential fatty acids produced conflicting results for canine atopic dermatitis [26,27,28,29,30] and studies failed to detect the orally delivered essential fatty acids in the skin [31,32]. These data suggest that the topical route is superior. All in all, skin barrier therapy in dogs is lacking and far behind human medicine, where products such as emollients are an important part of, for instance, atopic dermatitis therapy [33].

It is not known whether this spot-on could reach lesional sites on a dog when applied on a single site. Our study only assessed the effects of this spot-on on locations close to the spot-on site and not on distant sites. These sites were chosen to limit, as much as possible, the differences in the TEWL and SSH between locations on a dog [33]. This model also excluded inter-breed differences, as only laboratory beagle dogs were used. The investigated product has also been evaluated in clinical settings, on dogs suffering from atopic dermatitis, in combination with lokivetmab and a shampoo containing essential oils, fatty acids and sodium PCA. The treated group showed an increased cosmetic score and decreased pruritus and lesion scores (CADLI) compared with a control group that received only lokivetmab [16]. These data could indicate a distant spot-on diffusion or an effect from the shampoo. 

The mechanism of action of ATOP 7^®^ spot-on is only partially known. In a model of atopic dermatitis using a reconstructed canine epidermis, the active ingredients of ATOP 7^®^ spot-on seemed to lead to an improvement in the morphology of the epidermis, an increase in filaggrin expression and a decrease in IL-8 secretion [34]. These mechanisms are likely to involve essential fatty acids and ceramides or the numerous essential oils contained within the product. Indeed, turmeric has some beneficial effects on skin health [35,36,37], including its use in topical galenic form for the treatment of atopic dermatitis [38]. Additionally, wintergreen oil has antioxidant and anti-inflammatory properties [39,40,41,42,43,44].

This study has several limitations. The low number of dogs included in this study may have limited the statistical significance of our results, especially for SSH values. Another possible bias was the abdominal location of the control site, whereas the test sites were located on the dorsum. This abdominal location of the control site was deliberately chosen to minimize the impact of the evaluated spot-on treatment on the control site, allowing us to use each dog as its own control. Additionally, no significant differences in TEWL or SSH were detected between the test sites and the control site before tape stripping. Finally, this model does not directly translate to in vivo trials and only serves to experimentally investigate the effects of such products.

## 5. Conclusions

Weekly applications of ATOP 7^®^ spot-on protected the surrounding skin’s barrier function in a tape stripping model of repeated acute barrier disruption in dogs. The increase in TEWL and SSH after tape stripping was significantly lower after the first two consecutive weekly applications of the product. Therefore, ATOP 7^®^ spot-on may be an effective adjunctive treatment for diseases with skin barrier defects, such as canine atopic dermatitis, but further studies are needed to confirm this effect.

## Figures and Tables

**Figure 1 vetsci-09-00390-f001:**
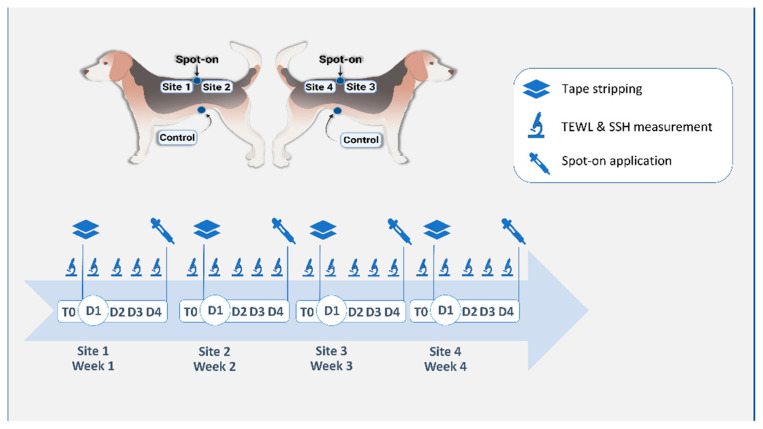
Schematic description of the study protocol. Four test sites were defined on the dorsum of the animal, as well as a spot-on site between the four test sites and a control site on the ventral abdomen. Tape stripping was performed on a single test site on the first day of the week corresponding to the site number (D1). Transepidermal water loss (TEWL) and skin surface hydration (SSH) were measured before (T0) and one hour after tape stripping (D1) and on three successive days after tape stripping (D2, D3, D4). Spot-on was applied on the defined site after measurements, on D4 of Weeks 1, 2 and 3.

**Figure 2 vetsci-09-00390-f002:**
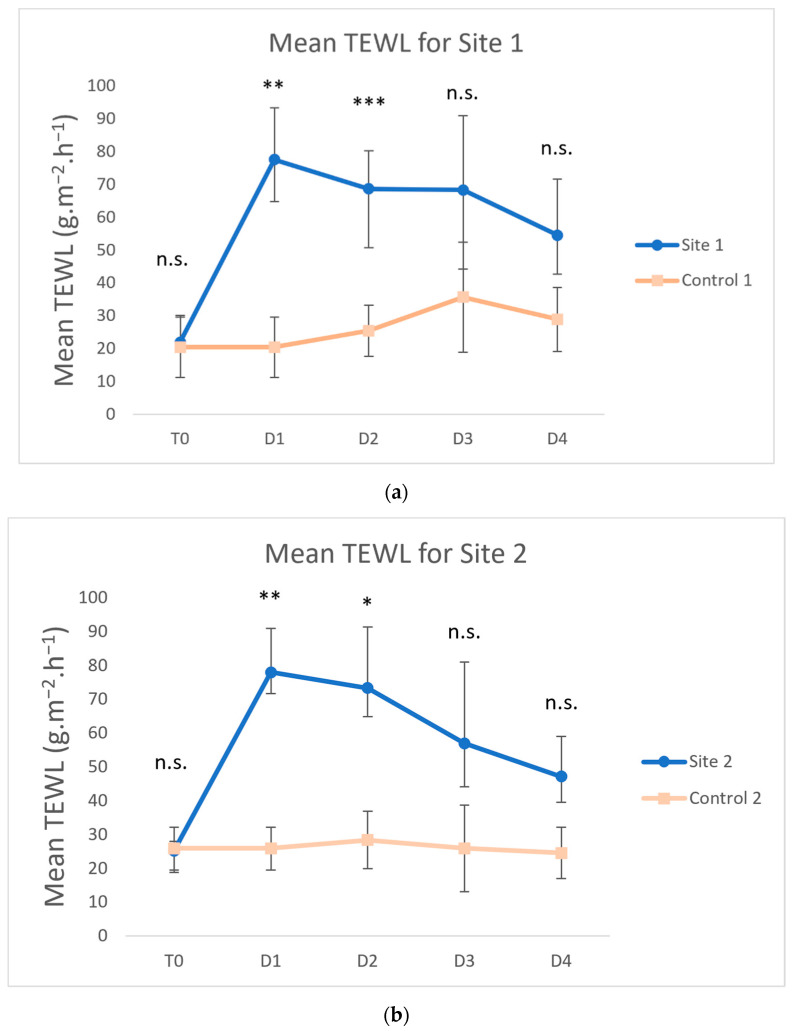
Means and standard deviations of transepidermal water loss (TEWL) for Site 1 (Week 1) (**a**), Site 2 (Week 2) (**b**), Site 3 (Week 3) (**c**) and Site 4 (Week 4) (**d**). Days were labeled as T0 (before tape stripping), D1 (one hour after tape stripping), D2 (24 h after tape stripping), D3 (48 h after tape stripping) and D4 (72 h after tape stripping). The values increased for test sites after tape stripping and decreased towards control values on the following days (*** *p* < 0.001; ** *p* < 0.01; * *p* < 0.05; n.s., not significantly different from control).

**Figure 3 vetsci-09-00390-f003:**
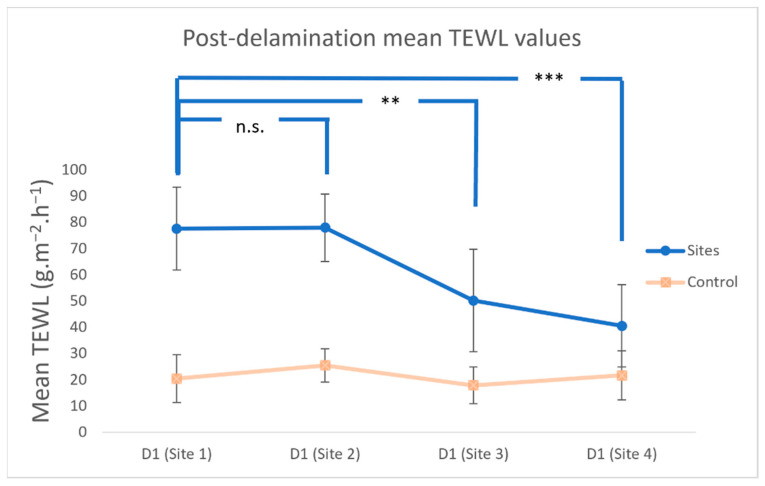
Means and standard deviations of transepidermal water loss (TEWL) after the same number of tape strips for each week. The evaluated spot-on was applied at the end of each week. (*** *p* < 0.001; ** *p* < 0.01; n.s., not significantly different from baseline).

**Figure 4 vetsci-09-00390-f004:**
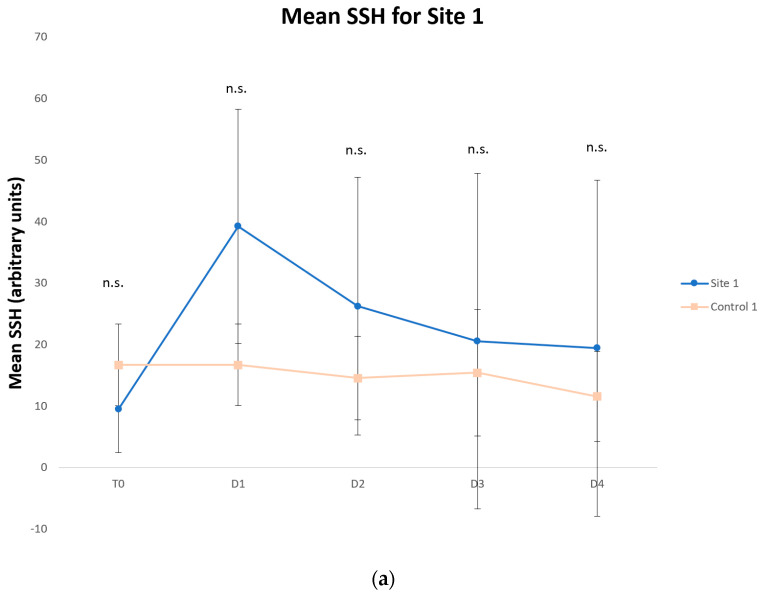
Means and standard deviations of skin surface hydration (SSH) for Site 1 (Week 1) (**a**), Site 2 (Week 2) (**b**), Site 3 (Week 3) (**c**) and Site 4 (Week 4) (**d**). Days were labeled as T0 (before tape stripping), D1 (one hour after tape stripping), D2 (24 h after tape stripping), D3 (48 h after tape stripping) and D4 (72 h after tape stripping). The values increased for test sites after tape stripping and decreased towards control values on the following days (n.s., not significantly different from control).

**Figure 5 vetsci-09-00390-f005:**
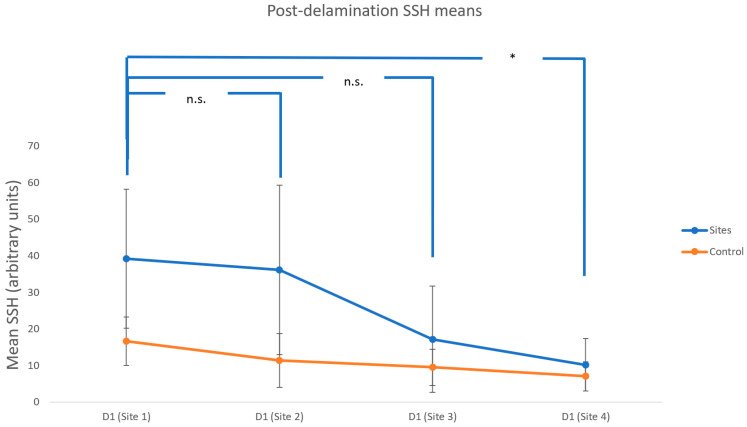
Mean post-delamination skin surface hydration (SSH). The evaluated spot-on was applied at the end of each week (* *p* < 0.05; n.s. not significantly different from baseline). Error bars represent standard deviation from the mean.

## Data Availability

Datasets are available at: Idée, Adrien (2022): ATOP 7 spot-on values.pzf. figshare. Dataset. https://doi.org/10.6084/m9.figshare.20102387.v1 (accessed on 23 June 2022).

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
