# Peer review of "Skin Barrier Reinforcement Effect Assessment of a Spot-on Based on Natural Ingredients in a Dog Model of Tape Stripping"

_vetsci, 2022, doi:10.3390/vetsci9080390_

Round 1

Reviewer 1 Report

The authors provide an exploratory study, conducted on a very small animal population, on the effect of a topical product (which has been investigated previously in a clinical setting) on trans-epithelial water loss and skin surface hydration.

Despite the topic is relevant and interesting, the paper is not very well written, it appears difficult to follow, especially in material and methods’ section and the discussion is poor. Introduction section does not provide sufficient background and does not include all relevant and updated references and, as a reflection of this, reference section (and the text in general, especially introduction) needs to be improved with more relevant paper.

Title: I advise to revise the title since the results presented were obtained in few dogs, so it seem to be too risky make such an assumption with the low number of dogs tested.

Lines 22-26: please provide references

Lines 32-38: please provide references for TEWL and SSH

Lines 46-47: authors state that “Even though the skin barrier has been extensively studied in dogs, few treatments 46 have been proven effective both experimentally and clinically” but no reference is provided – please provide those more relevant to the scope of the paper

Materials and methods: this section is not very clear and hard to follow (as an example, lines 79-81: what was the frequency of application of investigated product). Please completely rewrite, especially when explaining in detail the procedures of stripping in sites and days.

Lines 100-101: authors reported that “These data showed that our model of acute disruption of the skin barrier 100 worked as intended”, which appears to be a discussion of the result obtained. Thus, this phrase is not pertinent to result section, please move to discussion section

Lines 102-103: the same as above

Line 134: please remove “as expected” since this is a conclusion of the authors, not pertinent to result section, were the results obtained should be aseptically reported

Lines 153-154 this is a conclusion of the authors, not pertinent to result section, were the results obtained should be aseptically reported (again)

Figure 5: please detail how arbitrary units of SSH were determined.

Lines 190-192: the brand of the shampoo is irrelevant, it would have been interesting to read the composition instead

Conclusion section: this section needs to be completely rephrased since it appears excessive to attribute such properties to the investigated product in the trial herein presented

Author Response

Thank you for your revisions; please find the response within the attachment.

Reviewer 2 Report

This is a review of the manuscript entitled Skin barrier reinforcement effect of a spot-on based on natural ingredients in a dog model of tape stripping. This is an interesting study evaluating the effect of Atop 7® on an experimental model of canine skin barrier disruption. This study demonstrates new science, but like other similar studies, interpretation of the results should be cautious and clearly discussed within the limits of such study design. Here are some comments to authors.

Introduction:

I suggest to have the text edited by an English editor to improve the quality of the language.

This study wasn’t conducted in atopic dogs and wasn’t intended to find a topical therapy that specifically improves the skin barrier of atopic dogs. I suggest to the authors to delete the paragraph on canine atopic dermatitis or to cite this disease as example of skin barrier impairment, in which TEWL has been studied.

Line 36: ‘ As for TEWL, the deeper the measure is in the live epidermis, the higher the SSH value.’ Please clarify and detail this statement.

Line 50: ‘turmeric’ seems a better translation in that context since Curcuma refers to the plant itself.

Line 50: ‘wintergreen’ seems a better translation in that context since Gaultheria refers to the genus of the plant.

Materials and Methods:

Figure 1: in the legend of the figure, please define D1.

Lines 77-78: the number of tape strips is not mentioned. A shiny aspect of the skin seems to be quite subjective. In order to give readers enough information to compare with other studies and interpret the results (if 30 strips were performed on one dog and 15 on another, although the skin may look shiny on both dogs, the microscopic damage to the epidermis is not necessarily the same), I would appreciate if the authors would specify the number of strips per site/dog were performed.

Please specify in the M&M the clipping and the size of the blade used.

Results:

Lines 98-105: the results are presented by D1 to D4, but in Figure 1, only D1 is illustrated. For readers, it is not obvious that D1 to D4 are repeated for each week (week 1, week 2, etc.). I suggest clarifying the timeline on figure 1 and clearly defining in the legend each abbreviation illustrated on the figure.

Figures 2 and 4: I would suggest to the authors to clarify the legend of these figures. Actually, it is a little confusing for the readers to go from T0, D1, D2… to week 1, week 2, week 3… I would suggest to clearly indicate that T0, D1, D2, D3 and D4 are successive days in the same week (T0= day before tape stripping, D1 = day of tape stripping, etc.).

Discussion:

Lines 171-175: the authors specify that they used the AquaFlux, after mentioning intra-inter-observer variability with other devices. However, the authors do not mention why AquaFlux would be more reliable than other devices. It would be interesting if the authors would briefly mention why this device would overcomes challenges of both open and closed-chamber devices and if this technique has been validated.

Lines 182-183: restoration of the skin barrier is certainly an important aspect when treating atopic dermatitis, but I recommend to delete this sentence since it’s not directly aligned with this study. This study is not designed specifically for atopic dermatitis, although atopic dermatitis can be mentioned as an example of skin barrier dysfunction.

General comment for the discussion:

I would appreciate more discussion by the authors about limitations of this study.

·       There is no mention about the fact that SSH and TEWL can normally differ from site to site, which could have affected the comparisons between sites.

·       Moreover, the beneficial effect of this topical is demonstrated in this study in areas close to the spot-on application. The results obtained can’t be extrapolated to the whole skin surface of the dog.

·       The number of dogs is not only low, but some variations have also been reported between breeds.

·       Finally, there is no mention in the discussion of the complexity of cutaneous disease in a sick dog vs an experimental model that does not reflect exactly the normal disease setting. The results obtained in an experimental setting will not necessarily be the same in an animal with a naturally ongoing skin disease.

·       To help discuss the limitations, I suggest to add this reference: Won-Seok Oh, Tae-Ho Oh. Mapping of the dog skin based on biophysical measurements. Vet Dermatol 2010: 21 (4): 367-372.

Conclusions:

Considering the study design, I think that the authors should be careful to not ‘overinterpret’ the results. I think that the authors should add some nuances in their conclusions. The areas of the skin evaluated were close to where the spot-on was applied. In this study, there are no data regarding the effect of the Atop 7® spot-on on other areas of the skin. I think suggesting that this treatment can address atopic dermatitis, which is a complex disease even just in terms of skin barrier dysfunction, seems an ‘overinterpretation’ of the results. I would recommend to delete this sentence, especially, again, that the study was not designed to specifically evaluate this treatment in canine atopic dermatitis.

Author Response

Thank you for your revisions. Please find our response in the attachment.

Round 2

Reviewer 1 Report

I would congratulate with the authors for the revisions produced, which have made the manuscript acceptable in the present form